# Subacute Thyroiditis—Still a Diagnostic Challenge: Data from an Observational Study

**DOI:** 10.3390/ijerph19159388

**Published:** 2022-07-31

**Authors:** Taiba Zornitzki, Sorcha Mildiner, Tal Schiller, Alena Kirzhner, Viviana Ostrovsky, Hilla Knobler

**Affiliations:** 1Diabetes, Endocrinology and Metabolic Disease Institute, Kaplan Medical Center, The Faculty of Medicine, Hebrew University of Jerusalem, Rehovot 9190401, Israel; schillerta@gmail.com (T.S.); ide.elena@gmail.com (A.K.); viviana.ostrovsky@gmail.com (V.O.); hilla.knobler@gmail.com (H.K.); 2Internal Medicine Department, Kaplan Medical Center, Rehovot 9190401, Israel; sorcham@gmail.com

**Keywords:** subacute, thyroiditis, thyroid scan, quality of care

## Abstract

**Background:** Subacute thyroiditis (SAT) is a relatively common cause of thyroid disease. However, only a few studies evaluating SAT have been published in recent years with varying diagnostic criteria. We evaluate the clinical presentation and long-term outcome of isotope scan-confirmed SAT. **Methods:** A retrospective study of 38 patients with isotope scan-confirmed SAT was performed at a single isotope department. All patients were contacted for long-term follow-up. **Results:** The female/male ratio was 1.4:1, and mean age was 47 ± 14 years and 62 ± 12 years in women and men, respectively (*p* = 0.002). Almost half of the cases (42%) occurred during the summer. The most common symptoms were neck pain (74%) and weakness (61%). Palpitations, weight loss, heat intolerance, and sweating appeared in 50%, 42%, 21%, and 21%, respectively. Only half of the patients reported fever. TSH level was low in all patients, and mean FT4 and FT3 level were about twice the upper limit of normal range. Elevated CRP and ESR occurred in the majority (88%) of patients. The mean time period between the first clinic visit and performing thyroid function tests was 8 ± 7 days. One-third of the patients initially received a diagnosis of upper respiratory tract infection (URI). NSAIDs and steroids were prescribed to 47% and 8% of patients, respectively. Long-term follow-up of 33.5 months (range 9–52) revealed that 25% remained with subclinical or overt hypothyroidism. **Conclusions:** These data demonstrate that although SAT is a common entity, there is still a significant delay in diagnosis, and in a third of our patients, the initial diagnosis was URI, with 25% developing long-term hypothyroidism.

## 1. Introduction

Subacute thyroiditis (SAT), i.e., de Quervain’s disease, also known as granulomatous, giant cell or granulocytic thyroiditis, is the result of an inflammatory process of the thyroid gland with an unknown etiology and encountered in up to 5% of patients with thyroid illnesses [1]. Despite being a common disease, there are limited data on the incidence and clinical characteristics of SAT, due in part to a lack of strict diagnostic criteria [2,3,4,5,6,7,8,9]. Previous studies included mostly clinical symptoms such as local neck pain and tenderness, with abnormal thyroid function tests and inflammatory markers confirmed by either thyroid sonography, thyroid scan, or thyroid fine needle aspiration (FNA). In most studies, radioiodine uptake or a thyroid isotope scan were not mandatory criteria for SAT diagnosis, despite the high yield in the differential diagnosis of thyrotoxicosis. A thyroid isotope scan, can for example, effectively differentiate between Graves’ disease and SAT [4,5,6,7,8,9,10,11]. SAT is thought to have a viral origin and a genetic susceptibility [12].

The disease is characterized by three phases: 1. active thyrotoxic phase; 2. reactive hypothyroidism; 3. return to normal thyroid function. It is generally accepted that SAT is a self-limited disease with subsequent full thyroid function restoration. However, this commonly held notion has been challenged by a few previous studies that showed permanent hypothyroidism in 11–26% of patients [2,5,6,7]. Treatment is directed towards symptoms and varies according to disease phase and severity. Laboratory tests at the beginning of the disease show a typical pattern of suppressed thyroid-stimulating hormone (TSH), high free T4 (FT4) and free T3 (FT3) levels, and a high FT4:FT3 ratio that is supportive of thyroiditis and not of Graves’ disease [13]. Inflammatory markers such as erythrocyte sedimentation rate (ESR) and C-reactive protein (CRP) are commonly elevated. In some cases, the intense inflammatory process can lead to unnecessary investigations, and thyroiditis is still considered in the differential diagnosis of fever of unknown origin (FUO) [14,15]. Since SAT symptoms are not always specific, a diagnostic delay has been reported [8,16]. Nuclear scintigraphy in the initial stage of the disease reveals a characteristic absent radioisotope uptake (due to the release of preformed hormones from the damaged thyroid gland and TSH suppression), which confirms the diagnosis [10,11,17]. The aim of the present study was to evaluate the clinical presentation and long-term outcome of isotope-scan-confirmed SAT.

## 2. Methods

### 2.1. Patients and Study Design

A retrospective study was performed at Kaplan Medical Center, a university affiliated hospital. We reviewed records of all thyroid scans performed in the Isotope Department which provides services to both the hospital and the proximal community (a radius of about 40 km). The scans reviewed were from 1 January 2015 until 31 December 2018. In addition, we searched the electronic medical records (EMRs) of all patients with findings suggestive of thyroiditis by using two separate EMR systems: our institute’s EMR (Chameleon, Elad Solutions Ltd., Tel-Aviv, Israel) and an integrated hospital–community EMR (Ofek database system, Clalit Health Services, Israel). These data systems allow online access to community and hospital visits, laboratory and imaging data, and medications.

Inclusion criteria were typical technetium scan findings consistent with thyroiditis, including low or absent tracer uptake in the thyroid gland in the thyrotoxicosis phase [10,11,17], and age above 18 years. Exclusion criteria included postpartum state (occurring within 12 months from the end of pregnancy), thyroid malignancy, and exposure to iodine by either iodine-containing contrast material or by medications (within 12 months prior to scan). A total of 684 scans were conducted during the period from 1 January 2015 until 31 December 2018. Forty-seven (6.8%) thyroid scans fulfilled the criteria of thyroiditis. Nine patients were excluded: one case of postpartum thyroiditis, one case of metastatic carcinoma, and seven cases of amiodarone treatment. The study protocol was approved by the local Ethics Committee of Kaplan Medical Center, Rehovot, Israel, and was conducted in accordance with the Declaration of Helsinki. Informed consent to participate in the study was obtained from all subjects.

### 2.2. Assessment

By using the electronic data systems, the following data were obtained for all patients who fulfilled the SAT inclusion and exclusion criteria: 1. The clinical signs and symptoms. 2. Laboratory and imaging data, including: TSH, FT4, FT3 levels, ESR and CRP, blood count, liver enzymes, thyroid antibodies, and all further diagnostic investigations. 3. Medical history, including diagnoses made during the course of the disease, community clinic visits, hospitalizations, referrals, and attendance to emergency rooms or to outpatient clinics. In the second phase of the study, patients who fulfilled SAT criteria were interviewed by telephone questionnaire to evaluate long-term health status. Thirty-one responded to the questionnaire, and one patient died from an ischemic event nine months after thyroiditis (thyroid function was within normal limits one month before the cardiac event). Six patients could not recall the details regarding disease course. Latest thyroid function tests were obtained for all study participants.

The data are presented as mean and median for averages, percentages, standard deviation, and frequency. To compare continuous variables between groups, we used the non-parametric Mann–Whitney test. To examine the correlation between two non-continuous variables, we used Pearson and Chi-squared tests. Results were considered statistically significant when the *p*-value was <0.05. Data were analyzed using SPSS-21 software (IBM SPSS Statistics for Windows, Version 21.0., Armonk, NY, USA).

## 3. Results

Thirty-eight patients were included in the study. The female/male ratio was 1.4:1, with a mean age of 47 ± 14 years in women and 62 ± 12 years in men (*p* = 0.002). Almost half of the cases 16/38 (42%) occurred during the summer. The patients’ symptoms are shown in Figure 1. Twenty-eight (74%) presented with a sore or swollen throat, twenty-three (61%) complained of weakness, and nineteen (50%) complained of palpitations. Weight loss and heat intolerance were mostly reported by female patients, but this sex difference did not reach statistical significance (*p* = 0.08). Interestingly, only half of the patients, equaling 19 (50%), reported fever, and 3 patients presented with chest pain and/or diarrhea.

All patients underwent blood tests for thyroid function. All included patients were in the thyrotoxic phase of the disease during the time the technetium scintigraphy scan was performed. The mean time between the initial clinic visit and the referral for thyroid function test was eight days (range 1–48). Table 1 summarizes the laboratory data. Mean TSH level was 0.02 ± 0.06 mU/L, mean FT4 level was 40.6 ± 22.3 pmol/L, and mean FT3 level 14.1 ± 7.2 pmol/L. The T3/T4 ratio was 1:3, consistent with the diagnosis of thyroiditis [13]. The length of the hyperthyroid phase of the illness was available in 31 patients and was between 3 and 8 weeks in 27 patients (71%), 14 weeks in 2 patients, and 1 week in 2 patients. Elevated levels of thyroid peroxidase (TPO) and thyroglobulin (TG) antibodies were found in 13% and 24% of patients, respectively. Elevated ESR (mean 71 ± 38 mm/h) and/or an increased CRP (mean 6.0 ± 5.0 mg/dL) were found in more than two-thirds of the patients (71%). A full blood count revealed abnormality in 22 patients, and 53% had anemia of varying degree, with an additional decrease in hemoglobin during the period of inflammation in patients with pre-existing anemia. A quarter of the patients had leukocytosis (Table 1). Elevated liver enzymes were found in 28% of the patients, and increased alkaline phosphatase was the most common abnormality.

Twenty-five (66%) patients were diagnosed and treated only in the community. In the search for a diagnosis, six patients were hospitalized, five patients were evaluated in the emergency room, and two were evaluated in the hospital outpatient department. Reasons for hospitalization included: tachyarrhythmia (4 cases), prolonged diarrhea (1 case), and dyspnea (1 case). Thirty-one patients (82%) were referred to an ear, nose, and throat (ENT) clinic.

Figure 2 summarizes the laboratory tests and additional investigations performed in the study patients. Serological testing for viral infections, including Epstein–Barr virus (EBV), cytomegalovirus (CMV), and viral hepatitis, were conducted in 11 (29%) patients. Eleven (29%) patients underwent autoimmune diseases serology. Throat swabs and urine and blood cultures were performed in 15 (39%) patients. X-ray was performed in 17 (45%) patients: 16 chest X-rays and one cervical spine X-ray due to neck pain. One other patient underwent an abdominal CT scan (performed after the isotope scan) as part of the investigation of FUO. In three patients who presented with cardiac arrhythmia, echocardiography was performed. Twenty-nine patients (76%) underwent thyroid ultrasound, and among them, twenty-one (72%) had typical findings for thyroiditis: coarse echotexture and heterogeneous or nodular appearance. In three patients, an ultrasound found large nodules, which disappeared in a follow-up examination.

The exact time lag to achieve a diagnosis was available for 25 patients. Ten patients were diagnosed with SAT within two weeks, twelve within six weeks, and in three patients, diagnosis was reached more than six weeks after the first clinics’ visit.

The most common alternative diagnosis made by a primary physician was upper respiratory tract infection (URI). Two patients were diagnosed with Graves’ disease. A list of alternative diagnoses is shown in Figure 3.

Two-thirds of the patients were treated during the illness. The therapeutic decision regarding SAT was according to the judgment of the treating physician. The general policy was to start NSAID treatment and switch to steroids if there was insufficient response, side effects, or contraindications to NSAID treatment. A quarter of the patients (24%) received antibiotic treatment before the diagnosis of SAT was made. Twenty-two patients (58%) received beta blockers, 75% of males and 45% of females. Non-steroidal anti-inflammatory drugs (NSAID) were prescribed to 18 patients, more often to females, and steroids were prescribed to 7 patients. One patient received high-dose aspirin. Methimazole was prescribed to four patients but was discontinued one week before the radioisotope scan date in accordance with the nuclear medicine department policy.

Clinical course and long-term follow-up:

The median duration of follow-up in the study was 33.5 months (range 9–52). During the course of the disease, an average of 7.3 (median 7, range 3–14) thyroid tests were performed per patient. Twenty-four (63%) patients had a documented hypothyroid phase. Six patients received thyroxine treatment during the hypothyroid phase of the illness. Evaluation of SAT recurrence was determined by searching laboratory and clinic visit data and by the information derived from the long-term structured questionnaire. Based on these data, we did not find evidence of SAT recurrence.

Three-quarters of the patients had a complete recovery, five patients had a sequela of subclinical hypothyroidism, and four developed overt hypothyroidism and received thyroxine treatment. Hypothyroidism development was associated with NSAID treatment (*p* < 0.04). However, there was no association between development of hypothyroidism and the severity of thyrotoxicosis, inflammatory markers, gender, age, or steroid treatment.

## 4. Discussion

The current study included only patients who underwent a thyroid technetium scan, had results consistent with SAT, and lacked other conditions that can interfere with the scan results. We used isotope scanning with technetium, as it is an accurate non-invasive diagnostic tool for diagnosing SAT [10,11,17]. Our diagnostic approach differs from previous studies, in which diagnosis was based on a variety of inclusion criteria including clinical symptoms, laboratory findings, ultrasonography features, and biopsy findings, but neither a technetium scan nor iodine uptake findings were mandatory as diagnostic criteria of SAT [3,4,5,12]. Thus, the current study includes a more homogenous group of patients. This difference in design can explain some of the dissimilarities in patient symptoms between our study and previous studies. For example, while SAT is usually considered to be a painful inflammation of the thyroid gland, in our study, only about two-thirds suffered from sore or painful throat, compared to previous studies, in which sore throat and thyroid pain were used as inclusion criteria [4,7,8,9]. Furthermore, fever was found only in about half of the patients in the current study, in contrast to some previous data in which fever was reported in most patients [2,16]. Taken together, these data imply that the clinical picture of SAT depends to a large extent on the diagnostic criteria used.

Another different characteristic in our study was the relatively lower male to female ratio compared to previous studies [2,3,4,5]. Most cases in our study occurred during the summer, in agreement with the results of several previous studies [2,4,18].

In the present study, half of the patients had symptoms suggestive of thyrotoxicosis, such as palpitations, weight loss, heat intolerance, and excessive sweating. Such symptoms led in these patients to an early referral for thyroid function tests and consequently to an early diagnosis of SAT. In those lacking such symptoms, diagnosis was often delayed, and some of them experienced needless treatments or investigations, which even carried on for weeks, before establishing the diagnosis of SAT. For example, patients presenting with sore or swollen throat with or without fever were likely to be diagnosed with URI and treated with antibiotics. Notably, fever and elevated inflammatory markers such as ESR and CRP were not always present in our scan-diagnosed SAT patients, which also led to diagnostic delay. These data imply that concomitant fever and inflammatory markers in our study were not mandatory SAT criteria, in contrast to previous studies [4,7,8,9]. Our data demonstrate that SAT can present with any of the following clinical pictures: 1. symptoms of thyrotoxicosis with or without fever and/or inflammatory markers; 2. painful thyroid gland with or without fever and/or inflammatory markers; 3. prolonged fever, weight loss, and inflammatory markers.

The majority of patients were referred to an ENT specialist for further investigation. A substantial number of patients were hospitalized or referred to the emergency room. Many underwent unnecessary radiographic or serological testing for infectious or autoimmune etiologies. Due to an alternative diagnosis made at the initial stage, many of our patients received inappropriate treatment, mainly antibiotics, for a presumed URI or even an antithyroid drug for a supposed diagnosis of Graves’ disease. The delay in diagnosis, along with needless investigations and treatments, place an additional burden on both the patient and the healthcare system.

A commonly held notion is that most patients recover from SAT without impaired thyroid function. However, the findings of our study show that a quarter of the patients were found to have sub-clinical or overt hypothyroidism upon long-term follow-up, confirming previous data reporting hypothyroidism in 10–15% of patients with SAT [2,5,6,7]. The severity of the inflammation and the thyrotoxic phase of the disease were not related to the eventual outcome of the patients. There was also no association between the development of hypothyroidism and gender, age, or treatment modalities. There were no patients with recurrence of thyroiditis with follow-up of up to 4 years in our study. In previous studies, most recurrences were within the first years after the initial episode, with a wide range of 1.6% to 20% of patients [2,3,4,5,6,9,19].

Advantages and limitations of the study:

The main advantage of the current study is applying a strict inclusion criteria of isotope-scan-confirmed SAT. Another advantage is the use of a medical database that included both hospital and community healthcare information, enabling a comprehensive clinical evaluation of symptoms and management. We also conducted a long-term follow-up by using both medical records and a patient questionnaire. This approach provided detailed information regarding symptoms, alternative diagnoses raised, investigations, treatments provided, and long-term follow-up.

The main limitation is that SAT is a self-limiting disease that can resolve without being diagnosed by primary care physicians. Therefore, mild cases may not be referred to further evaluation, including a thyroid scan. A potential bias of our study is therefore that the patients included may had more severe SAT or an atypical presentation. Another limitation of the study is the small sample size of the cohort.

## 5. Conclusions

In our isotope-scan-confirmed SAT study, localized symptoms of sore throat or thyroid tenderness, fever, and elevated inflammatory markers were presented only in half to about two-thirds of the patients. Our data demonstrate that in about a quarter of patients, there were no fever or inflammatory markers. Therefore, SAT is a more heterogenous disease, which also includes atypical presentations. The heterogenous clinical picture led in many patients to an erroneous diagnosis and unnecessary investigations and treatment. In addition, our data demonstrate that with long-term follow-up, a quarter of our patients remained with subclinical or overt hypothyroidism. Thus, we suggest long-term follow-up of the thyroid function test in patients with SAT.

## Figures and Tables

**Figure 1 ijerph-19-09388-f001:**
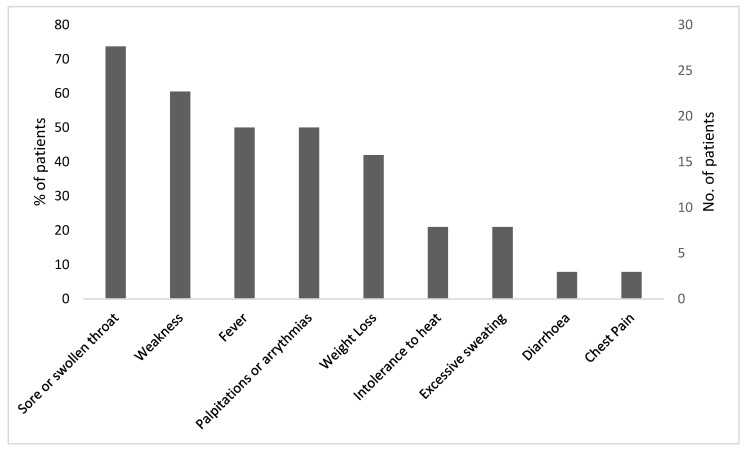
Clinical symptoms in patients with SAT (*n* = 38).

**Figure 2 ijerph-19-09388-f002:**
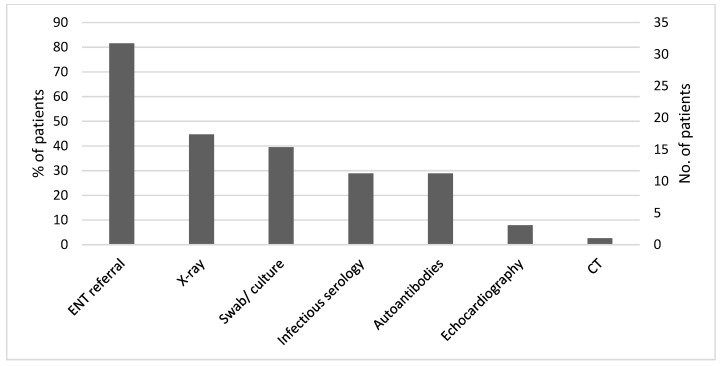
Investigations performed in search of an alternative diagnosis (*n* = 38).

**Figure 3 ijerph-19-09388-f003:**
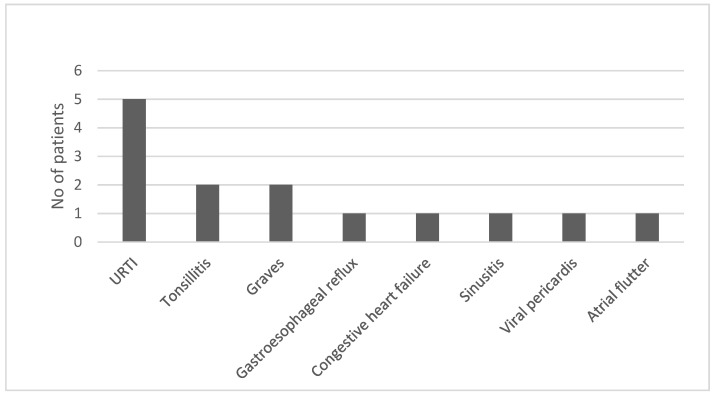
Initial alternative diagnoses (*n* = 38).

**Table 1 ijerph-19-09388-t001:** Initial Laboratory results for SAT patients during active thyrotoxic phase (*n* = 38).

Variable	Normal Range	Mean ± SD	Median (Range)
Minimum TSH level (mIU/L)	0.55–4.78	0.02 ± 0.06	0.01 (0.008–0.1)
FT4 (pmol/L) at time of scan	10.0–20.0	40.6 ± 22.3	33.7 (12.1–128.3)
FT3 (pmol/L) at time of scan	3.5–6.5	14.1 ± 7.2	11.7 (6.8–31.0)
TPO-ab (IU/mL)	0–35	49 ± 180	0 (0–998)
TG-ab (IU/mL)	10–40	123 ± 457	0 (0–2454)
Leukocytosis (10^3^/mm^3^)	4.5–11.5	12.6 ± 1.0	12.3 (11.6–14.6)
Hb (g/dL)	12–16	12.3 ± 2.0	12.5 (7.3–14.8)
CRP (mg/dL)	<0.5	6.0 ± 5.0	6.0 (0–16)
ESR (mm/h)	<22	71 ± 38.0	73 (2–125)

Abbreviations: SD—standard deviation; TSH—thyroid-stimulating hormone; TPO—thyroid peroxidase antibodies; TG—thyroglobulin antibodies; Hb—hemoglobin; CRP—C-reactive protein; ESR—erythrocyte sedimentation rate.

## Data Availability

The data supporting the findings of this study are available within the article.

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
