# Peer review of "Subacute Thyroiditis—Still a Diagnostic Challenge: Data from an Observational Study"

_ijerph, 2022, doi:10.3390/ijerph19159388_

Round 1

Reviewer 1 Report

The legend of your table must be improoved , but the papaer are very intetresting  and well done 

Author Response

We thank the editor and reviewers for their important and thoughtful comments. Please see below, our response to each comment.

The legend of your table must be improoved , but the papaer are very intetresting  and well done 

Response 1: The legend of table 1. was changed accordingly: Initial laboratory results of SAT patients during active thyrotoxicosis phase (n=38)

Reviewer 2 Report

Zornitzki and colleagues reported a small series of patients with SAT diagnosed by isotope-scan. While the topic is still matter of debate, there are larger case series in literature. The selection criteria could represent a novelty of the study, but there are some points that need to be addressed.

11)      How many scans were performed in this time-period upon suspicion of thyroiditis? More data about the usefulness of isotope-scan in the differential diagnosis of SAT should be provided.

22)      Please add more details about the follow-up evaluation. How frequently were thyroid function tests repeated? Which markers were evaluated? How were treatments adjusted?

33)      Is it possible that follow-up assessment via questionnaire led to underestimate recurrence?

44)      Please add more details about the reasons driving the treatment of patients (NSAIDs vs steroids).

55)      The outcome of patients should be contextualized in time. Please add the percentage of patients that recovered an euthyroid status after 1 year and at the end of follow up. This would give more information about early and late hypothyroidism after SAT. In addition, which was the median follow-up?

66)      Sample size is indeed a limitation of the study. This should be acknowledged.

77)      Some authors excluded patients undergoing levothyroxine treatment from permanent hypothyroidism assessment. Please comment your choice to keep them.

88)      The authors state that “fever and inflammatory markers are not mandatory SAT criteria” (lines 207-208). Yet, increased inflammatory markers and fever were reported in 71 and 50% patients of this cohort respectively. No data are provided about their combination to support this statement, nor information about the performance of alternative diagnostic tools is provided. In my opinion, these data (i.e., inflammatory markers and fever) are easy to collect and useful combined with other clinical information and diagnostic testing, though they are non-specific. Please comment and maybe soften this statement.

Author Response

We thank the editor and reviewers for their important and thoughtful comments. Please see below, our response to each comment.

REVIEWER 2.

11)  How many scans were performed in this time-period upon suspicion of thyroiditis? More data about the usefulness of isotope-scan in the differential diagnosis of SAT should be provided.

       Response 1: 684 thyroid scans were conducted in the Isotope Department of our hospital during the period of 01/01/2015-31/12/2018. Forty-seven (6.8%) out of 684 thyroid scans fulfilled the criteria of thyroiditis with typical findings of low or absent tracer uptake in the thyroid gland at time of the thyrotoxicosis phase. We added this to the Methods section (lines 82-84, 87-88). A sentence about the usefulness of isotope diagnosis in differential diagnosis of SAT was added to the Introduction section (lines 50-51).

22)  Please add more details about the follow-up evaluation. How frequently were thyroid function tests repeated? Which markers were evaluated? How were treatments adjusted (Sorcha)?

Response 2: The median duration of follow-up in the study was 33.5 month (range 9-52). During the course of the follow-up period, an average of 7.3 (median 7, range 3-14) thyroid tests were performed per patient. We added this to the Abstract and Results section (lines 31, 189-191). Additional laboratory tests included inflammatory markers including ESR, CRP and total blood count, liver function, renal function, antibodies to autoimmune disease and viral disease, throat swabs, urine and blood cultures.

The therapeutic decision regarding SAT thyroiditis was according to the judgment of the threating physician. The general policy is to start NSAID treatment and switch to steroids if there is insufficient response, or side effects, or contraindications to NSAID treatment. We added this to the Results section (lines 178-181).

33)  Is it possible that follow-up assessment via questionnaire led to underestimate recurrence- Sorcha? Response 3: We cannot eliminate the possibility of underestimation in some patients of unrecognized recurrence. However, we conducted relatively long follow-up of 33.5 months (range 9- 52). Thyroid function tests were repeated on average 7.3 times (median 7, range 3-14) per patient.

44) Please add more details about the reasons driving the treatment of patients (NSAIDs vs steroids).

The therapeutic decision regarding SAT thyroiditis was according to the judgment of the threating physician. The general policy is to start NSAID treatment and switch to steroids if there is insufficient response, or side effects, or contraindications to NSAID treatment. We added this to the Results section (lines 178-181).

55) The outcome of patients should be contextualized in time. Please add the percentage of patients that recovered an euthyroid status after 1 year and at the end of follow up. This would give more information about early and late hypothyroidism after SAT. In addition, which was the median follow-up (Sorcha)?

       Response 5: Thirty-two (84%) patients had normal thyroid function towards the end of the first follow-up year and twenty-nine (76%) at the end of the follow-up period.

66)   Sample size is indeed a limitation of the study. This should be acknowledged.

         Response 6: We added study small sample size as a limitation in the Discussion section (lines 266-267).

77)  Some authors excluded patients undergoing levothyroxine treatment from permanent hypothyroidism assessment. Please comment your choice to keep them.

       Response 7. The reason to include them as permanent hypothyroidism was that with long term follow-up, they continued to be treated with levothyroxine.

88)  The authors state that “fever and inflammatory markers are not mandatory SAT criteria” (lines 207-208). Yet, increased inflammatory markers and fever were reported in 71 and 50% patients of this cohort respectively. No data are provided about their combination to support this statement, nor information about the performance of alternative diagnostic tools is provided. In my opinion, these data (i.e., inflammatory markers and fever) are easy to collect and useful combined with other clinical information and diagnostic testing, though they are non-specific. Please comment and maybe soften this statement.

       Response 8: Four of the patients had neither fever nor increased inflammatory markers. We changed the sentence “fever and inflammatory markers are not mandatory SAT criteria” according to reviewer suggestion: Concomitant fever and inflammatory markers in our study were not mandatory SAT criteria in contrast to previous studies (lines 231-232)

Reviewer 3 Report

Review in the attachment

Author Response

We thank the editor and reviewers for their important and thoughtful comments. Please see below, our response to each comment.

REVIEWER 3.

I propose to add to the title: Retrospective observational study

Response 1. We change the name of the study according to the reviewer suggestion: Subacute thyroiditis - still a diagnostic challenge: data from an observational study (line2-3).

11- The abstract should be written in accordance with the chapters of the work (250 words)

Response 2: A structured abstract of 250 words, was written in accordance with the above comment.

12- It is not a common inflammation

Response 3: According to reviewer suggestion, we changed the sentence: “SAT is a relatively common cause of thyroid disease” (line 14).

14- what does "consecutive patients" mean?

Response 4: We deleted the word “consecutive” (line 19) in the abstract.

28- What does an unusual presentation mean?

Response 5. We deleted the “atypical presentation” (line 38) from the Key words section.

31 - I propose at the beginning of the introduction: Subacute thyroiditis, i.e. de Quervain's disease, also known as granulomatous, giant cell or granulocytic thyroiditis, is the result of an inflammatory process of the thyroid gland with an unknown etiology

Response 6: We added the recommended sentence to the Introduction section (lines 41-43).

57- How far away is your research?

Response 7: The median follow-up time in the study was 33.5 month (range 9-52). We added this to the Abstract and Results section (lines 31, 189-191).

68- In what period of the disease were patients included in the study using technetium scintigraphy 73- What are the criteria?

Response 8: All included patients were in the thyrotoxic phase of the disease during the time the technetium scintigraphy scan was performed. We added this to the Results section (lines 129-131). Inclusion criteria to the study involvement were: typical technetium scan findings consistent with thyroiditis and age above 18 years during period 01/01/2015-31/12/2022. Exclusion criteria included postpartum state (occurring within 12 months from the end of pregnancy), thyroid malignancy, and exposure to iodine by either iodine-containing contrast material or by medications (within 12 months prior to scan) (lines 82-87).

59- Lack of precise description of patients - total number of patients, the number of meeting and failing criteria, how many patients dropped out of observation, and how many were ultimately followed up. 82- what imaging examinations were used?

Response 9: 684 thyroid scans were conducted in Isotope Department of our hospital during the period of 01/01/2015-31/12/2018. We added this to the Methods section (lines 87-88). Inclusion criteria to the study involvement were typical technetium scan findings consistent with thyroiditis and age above 18 years. Forty-seven (6.8%) out of 684 thyroid scans fulfilled the criteria of thyroiditis with typical findings of low or absent tracer uptake in the thyroid gland in the thyrotoxicosis phase. Exclusion criteria included postpartum state (occurring within 12 months from the end of pregnancy), thyroid malignancy, and exposure to iodine by either iodine-containing contrast material or by medications (within 12 months prior to scan). Nine patients were excluded: one case of postpartum thyroiditis, one case of metastatic carcinoma and seven cases of amiodarone treatment. We examined clinical and laboratory data of all patients, who met inclusion and exclusion criteria, from hospital and HMO files (82-92). In parallel, all patients who fulfilled inclusion criteria were interviewed by telephone questionnaire to evaluate long-term health status. Thirty-one patients responded to the questionnaire, one patient died from an ischemic event nine month after thyroiditis (thyroid function was within normal limits one months before the cardiac event) (lines 102-104). Six patients could not recall the details regarding disease course. This was added to the Assessment section (lines 104-105).

X-ray was performed in 17 (45%) patients: 16 chest X-rays, and one cervical spine X-ray due to neck pain. One other patient underwent an abdominal CT scan, (performed after the isotope scan), as part of the investigation of FUO. In three patients who presented with cardiac arrhythmia, an echocardiography was performed. Twenty-nine patients (76%) underwent thyroid ultrasound (lines 160-164).

86- when did the second phase of the research (follow up) start?

Response 10: The second phase of the research was conducted from 01/01/2019 until 31/06/2019.

107- in the description we do not repeat the information contained in the tables and figures. We avoid repetitions, we provide the highest and the lowest statistically significant data.

Response 11: We shortened Results section in according to reviewer suggestion (lines 120-122).

108- How many patients were analyzed

Response 12: We added number of patients in the figure legends (125,145, 168, 176).

167- How long was the observation?

Response 13: The median follow-up time in the study was 33.5 month (range 9-52). We added this to the Abstract and Results section (lines 31, 189-191).

172- how much time passed from diagnosis to complete recovery

Response 14: The average time between diagnosis and recovery was 4.3 months (median 4, range 2-12).

175- Where Is it shown. (Multivariate Analysis)?

Response 15: We conducted a univariate analysis to evaluate association between the hypothyroidism development and the severity of thyrotoxicosis, inflammatory markers, gender, age and steroid treatment. The only one that was found to be associated with hypothyroidism development was NSAID treatment (p<0.04), and therefore multivariate analysis was not done. The association between NSAID treatment and hypothyroidism was added to the Results section (lines 198-201).

226- Where is no relationship shown?

Response 16: We conducted a univariate analysis to evaluate association between the hypothyroidism development and the severity of thyrotoxicosis, inflammatory markers, gender, age and steroid treatment. The only one that was found to be associated with hypothyroidism development was NSAID treatment (p<0.04), and therefore multivariate analysis was not done. The association between NSAID treatment and hypothyroidism was added to the Results section (lines 198-201).

243- Conclusions are not a summary, they should be in the form of short statements resulting from the work and correspond to the set goal. Conclusions need to be thoroughly redrafted and written in the present tense

Response 17: We changed conclusion section in according to reviewer suggestion.

Round 2

Reviewer 2 Report

I have no further comments.